# The Application of Long-Read Sequencing to Cancer

**DOI:** 10.3390/cancers16071275

**Published:** 2024-03-25

**Authors:** Luca Ermini, Patrick Driguez

**Affiliations:** 1NORLUX Neuro-Oncology Laboratory, Department of Cancer Research, Luxembourg Institute of Health, L-1210 Luxembourg, Luxembourg; 2Bioscience Core Lab, King Abdullah University of Science and Technology, Thuwal 23955-6900, Saudi Arabia

**Keywords:** third-generation sequencing, short reads, long reads, cancer, precision oncology

## Abstract

**Simple Summary:**

Cancer is a complex disease caused by a slew of genetic mutations discovered through advances in sequencing technologies such as next-generation sequencing. While this technology has been useful, it can only retrieve genomic information through short reads or sequences, which is a limitation. A new sequencing technology known as third-generation sequencing overcomes this limitation by using much longer reads. This is a game changer for cancer research, diagnosis and treatment. Third-generation sequencing enables the decipherment of complex genomic rearrangements, resulting in a better understanding of how cancer develops, as well as the examination of the entire transcriptome, revealing isoforms that could be used in diagnostics or treatment. Third-generation sequencing enhances cancer genome assembly, detects epigenetic changes, and can provide a comprehensive picture of a patient’s specific cancer aberrations. This has the potential to lead to more effective treatments with fewer adverse effects. This review provides a rigorous scientific analysis of the advantages and limitations of third-generation sequencing, emphasizing its potential for the future of cancer research and personalized medicine. Although this is still a developing technology, it has enormous potential for research and clinical applications, ultimately leading to improved cancer diagnosis and treatment.

**Abstract:**

Cancer is a multifaceted disease arising from numerous genomic aberrations that have been identified as a result of advancements in sequencing technologies. While next-generation sequencing (NGS), which uses short reads, has transformed cancer research and diagnostics, it is limited by read length. Third-generation sequencing (TGS), led by the Pacific Biosciences and Oxford Nanopore Technologies platforms, employs long-read sequences, which have marked a paradigm shift in cancer research. Cancer genomes often harbour complex events, and TGS, with its ability to span large genomic regions, has facilitated their characterisation, providing a better understanding of how complex rearrangements affect cancer initiation and progression. TGS has also characterised the entire transcriptome of various cancers, revealing cancer-associated isoforms that could serve as biomarkers or therapeutic targets. Furthermore, TGS has advanced cancer research by improving genome assemblies, detecting complex variants, and providing a more complete picture of transcriptomes and epigenomes. This review focuses on TGS and its growing role in cancer research. We investigate its advantages and limitations, providing a rigorous scientific analysis of its use in detecting previously hidden aberrations missed by NGS. This promising technology holds immense potential for both research and clinical applications, with far-reaching implications for cancer diagnosis and treatment.

## 1. Introduction

Cancer is a complex disease characterised by the uncontrolled proliferation of tumorigenic transformed cells subject to evolution by natural selection [1]. Cancer cells can rapidly proliferate within a tissue, spread outside of normal regulatory boundaries, invade neighbouring tissues, and even colonise distant sites [2]. This progression is modelled by an evolutionary process in which single cells grow as a result of interactions between single cells and the local microenvironment. Cancer cells undergo genotypic and phenotypic changes, the majority of which are driven by a wide range of genetic alterations, allowing them to improve cellular fitness and overcome environmental and treatment constraints. Cancer genomes are shaped by a variety of somatic alterations, including single nucleotide variants (SNV), short insertion/deletion (indels), structural (SV) and copy number variants, and very complex rearrangements, like chromoanagenesis (chromoplexy, chromoanasynthesis, and chromothripsis) and breakage–fusion–bridge cycles [3]. The identification of those somatic variants is a critical issue in cancer, and significant resources and effort have been made to report the genomic statuses specific to each cancer subtype and to compile a comprehensive catalogue of cancer somatic mutations [4,5]. Over the past decade, somatic mutations have been successfully identified using a short-read sequencing technology called next-generation sequencing (NGS), but due to inherent technological limitations, our understanding of somatic mutations in cancer is far from complete. NGS studies have mostly found point mutations, like SNVs and indels because short reads are best suited for mutations of a single base or short fragment length. With reads that are only a few hundred bases long, it is challenging to detect and accurately identify complex genomic alterations, like structural and copy-number variants, or mutations in repetitive regions. The development of new sequencing technologies based on long-read sequencing has ushered in the era of third-generation sequencing (TGS), also known as long-read sequencing (LRS), [6] which produces sequencing reads that are longer than NGS systems (Table 1). Reads lengths in excess of 10,000 bp are typical in TGS but ultra-long reads can also be achieved; a sequence read of 1.04 Mb was generated for a human chromosome [7], and the longest read ever reported in scientific literature is close to 2.3 Mb [8]. This technological advancement has made it possible to detect complex rearrangements, directly explore epigenetic modifications, characterise the entire transcriptome, and gain a better understanding of cancer genomics, all of which have the potential to significantly impact clinical and therapeutic approaches to cancer.

In this review, we describe third-generation technologies as they currently stand and introduce long-reads for cancer research and diagnosis, which could completely transform the fields of cancer biology and medicine.

## 2. The Promise and Limits of NGS in Cancer Research

Short-read next-generation sequencing refers to the next advancement of sequencing technologies after traditional first-generation Sanger sequencing. This technology was first introduced in the mid-2000s, ushering in a new era of scientific research. A key feature of short-read technologies is the parallel sequencing of short clonally amplified DNA molecules. Read lengths, while still shorter than Sanger sequencing, are a few hundred bases. But, the number of reads can be in the billions, and the single base accuracy is approximately 99.9% [9]. Since its introduction, the use of NGS in cancer research has grown rapidly due to the massive generation of highly accurate reads, which allows for the profiling of cancer genomes, identification of new mutations, prediction of neoantigens, detection of epigenetic modifications, and tracking cancer evolution [3,10,11]. Furthermore, NGS has been used to characterise the cancer transcriptome using RNA-seq, providing information on gene expression as well as detecting alternative splicing and fusion events [12]. NGS is also used in liquid biopsies, which identify circulating cancer cells, cell-free DNA (cfDNA), and circulating non-coding RNAs in blood or other bodily fluids [13], whereas single-cell sequencing via NGS is used to evaluate cancer heterogeneity and identify different cell types [14].

Despite the numerous scientific cancer breakthroughs using NGS, this sequencing technology seems to have reached its limit and new approaches are required. For example, the short-read length is one of the most evident limitations of NGS. Despite the generation of billions of short reads, reads of a few hundred bases in length are not optimal for genome assembly in the context of a complex genome, such as that of cancer. Short reads struggle with structural variations or even low-complexity regions when reassembling data over long stretches of DNA; large complex genomic rearrangements (>5 Mb) are even more challenging. However, for cancer SV detection, recent algorithm improvements show promise for NGS pipelines [15].

To boost the signal, NGS platforms use clonal amplification of DNA templates, and unlike TGS, cannot directly detect nucleotide changes at the single-molecule level. This is a significant limitation because the template amplification process can cause spurious mutations, or other technical artefacts, that can masquerade as sequence variants. In addition, template amplification introduces biases that under-represent AT-rich and GC-rich regions in target sequences [16]. Transcriptomes based on NGS are an excellent example of this limitation. Typically, RNA samples are first fragmented, then reverse transcribed, and then PCR amplified before the complementary DNA (cDNA) fragments are sequenced in a high throughput manner. This complex process introduces biases in the sequences by increasing base incorporation errors in individual molecules and by underrepresenting bases in regions with high or low GC contents [17].

Another limitation of NGS is its inability to directly sequence RNA or DNA, and NGS platforms are often blind to nucleotide modifications. To detect transcriptome-wide RNA modifications and epigenetic methylations, NGS must use indirect methods, which frequently fall short in identifying the underlying causes of the modification at a given site and in providing quantitative estimates of those changes [18].

What is the future of NGS then, given those flaws? Fortunately, we already have other sequencing technologies at our disposal, such as long-read third-generation sequencing, which offers a variety of approaches for analysing genomes and transcriptomes. A summary of long-read and short-read methods, as well as the advantages of each technology, is presented in Figure 1. Third-generation sequencing has enormous potential, which is only now becoming clear, and will improve our understanding of cancer biology. These state-of-the-art long-read sequencing techniques will be outlined in the following section.

## 3. Long Read Sequencing

### 3.1. Technology Background

Third-generation sequencing is the latest iteration of sequencing technology. Unlike NGS, TGS allows the sequencing of single molecules at lengths of up to tens of thousands of nucleotide bases, and even as long as megabases. The major TGS platforms are Oxford Nanopore Technologies (ONT) and Pacific Biosciences (PacBio).

ONT nucleotide basecalling is achieved when a single strand of a DNA or an RNA molecule is pulled through a protein nanopore, embedded in a synthetic membrane, by an electric potential. As the nucleic acid strand passes through the nanopore there is a change in current, or signal, that has a different profile for each of the nucleotide bases, enabling basecalling of the fragment. There is no limit to the length of nucleic acids that can be sequenced and megabase reads are common; the longest read of 4.2 Mb was reached in an internal test [19]. In addition, some modifications to native RNA and DNA bases cause a specific current change when pulled through the nanopore, allowing for direct detection. The number of modified bases that can be potentially detected in DNA and RNA is only limited by the training of the basecalling algorithm and having a distinct electrical signal associated with the modified base. Currently, different base modifications, such as 5-methylcytosine (5mC), 5-Hydroxymethylcytosine (5hmC), 6-methyladenosine (6mA) and Bromodeoxyuridine (BrdU), are detectable in DNA and N6-methyladenosine (m6A) in RNA [20].

PacBio uses sequencing-by-synthesis technology to measure the polymerase base incorporation in individual DNA molecules, between 1000 and 20,000 nucleotides in length, held in microwells (zero mode waveguides, ZMW). As the bound polymerase incorporates a fluorescently tagged nucleotide, the event is optically detected within each ZMW. The double-stranded DNA fragment has hairpin adapters at the ends that enable multiple passes over the same template using rolling circle amplification. Each pass, called a subread, can be compared and used to create a highly accurate consensus read of the fragment [21]. Methylation in DNA can cause a slight detectable delay during base extension (interpulse duration, IPD) in sequencing, specifically the 5-methyl-cytosine in CpG motifs are part of the standard PacBio basecalling pipeline [22,23].

A significant early limitation of long-read sequencing was the much lower base accuracy compared to NGS [24]. Early iterations of nanopore sequencing had an error rate as high as 30–40% [25], with subsequent improvements in accuracy [26]. The latest ONT chemistry, flow cells and basecalling algorithms have improved the base accuracy to >99% (quality score: Q20) and even up to 99.9% (Q30) [27], approaching the accuracy of NGS. Similarly, PacBio sequencing suffered from a significant error rate when it was first introduced. Approximately one in every ten bases was incorrectly identified, resulting in an error rate of 8–15% in continuous long reads, an earlier PacBio sequencing technology capable of sequencing long DNA fragments (typically >30 kb). The introduction of a new PacBio sequencing technology known as high fidelity (HiFi), which sequences a shortened DNA template (10–20 kb in length) multiple times, has significantly improved sequencing accuracy. A template sequenced four times is estimated to have 99% accuracy (Q20), while ten passes result in 99.9% accuracy (Q30) [28].

### 3.2. Advantages of TGS

One of the major advantages of TGS is that, unlike NGS, there is no amplification step during sequencing. As a result, both PacBio and ONT sequence individual native DNA fragments without potential amplification biases, and, moreover, base modifications are directly read. A notable aspect in the field of cancer research is that TGS platforms have the capability to identify 5-methylcytosine in DNA without the requirements of cumbersome bisulfite conversion protocols. In addition, ONT is able to detect 5mC and 6mA in DNA directly; however, ONT and the research community are independently developing methods to detect other DNA modifications [29]. Using long-read sequencing, primarily ONT ultralong reads, a complete and gapless (telomere-to-telomere) epigenome has been created, further enabling the study of epigenetic changes in cancer, including in previously unreachable parts of the genome [30]. Furthermore, via direct RNA sequencing, ONT is able to detect modifications in RNA molecules [31].

Nanopore platforms are highly scalable with real-time sequencing information, enabling portability and rapid on-site analysis of long DNA or RNA fragments. This is a distinct and significant advantage that can be leveraged in clinical settings where diagnosis can be of critical importance [32]. Using a high throughput ONT sequencer and parallel sequencing across multiple flow cells, it was possible to diagnose disease-causing genetic variants in less than 8 h for two critically ill patients [33].

An important advantage of TGS is the capacity to generate full-length mRNA transcripts, thereby providing complete information about alternative splicing events, isoforms, alternative polyadenylation, and gene fusions [34,35,36]. Despite the very high accuracy of PacBio mRNA sequencing (long-read isoform sequencing, or Iso-seq), on par with NGS, one limitation has been the lower number of reads per sequencing cell compared to NGS and ONT platforms. To overcome this, PacBio has implemented a new library protocol that concatenates multiple full-length transcripts on one sequencing molecule. This has increased the number of reads per sequencing cell for single cell Iso-Seq to 80–100 million and bulk Iso-Seq to 30–40 million, enabling full-length mRNA sequencing data to be used for applications that were previously restricted to NGS data, such as cell population clustering and differential analysis [37,38].

## 4. The Long-Read Approach in Cancer

The human genome contains many regions of varying complexity, some of which are relevant to cancer. Low-complexity tandem repeats, pseudogenes, regions with a high GC content, and regions with a high copy number variation are some examples [39]. Various consortia [5,40] have used short-read sequencing technology to sequence, analyse, and report genomic profiles for different types of cancer. Most of their studies were able to confidently detect point mutations, such as single-nucleotide variants and short indels. Some more complex genetic changes, like structural variants, can be harder to find with short-read technology. This issue is addressed by third-generation sequencing, which uses long reads to span over the complicated parts of the genome, including cancer genomes. Figure 2 provides an overview of how TGS is used to study cancer-related topics.

### 4.1. Cancer Genomes with Long Reads

#### 4.1.1. Identify and Phase Single Nucleotide Variants

Single nucleotide variants are the most common type of somatic variants and have gained increased interest due to their involvement in cancer progression [4]. These variants can arise in the DNA of a single cell and, through subsequent clonal expansion, lead to somatic clonal heterogeneity fuelling clonal evolution and cancer progression. The widespread use of high throughput short-read sequencing has characterised many SNVs. However, this technology necessitates PCR, which restricts SNV detection to regions that can be amplified, and short-read lengths pose challenges in resolving phasing. Phasing mutations in cancer provide valuable insights into patients’ specific genetic backgrounds, allowing for the development of personalised treatment strategies based on their unique genomic profiles. By utilising the specific genetic profiles of individual cancer patients, clinicians can make informed treatment decisions to optimise therapy and minimise drug side effects, thereby enhancing patient care. An example of how mutation phasing can be used in cancer treatment is in the case of epidermal growth factor receptor (*EGFR*) mutations in lung cancer. The *EGFR* gene is susceptible to two missense mutations: T790M and L858R. Lung cancer cells with only the L858R mutation are usually responsive to tyrosine kinase inhibitor drugs (TKIs), but when T790M is present, most often in the cis position, it confers acquired resistance. The presence of another mutation C797S in trans with T790M causes resistance to third-generation TKIs, but combination therapy with both first- and third-generation inhibitors showed sensitivity [41].

One of the advantages of long-read sequencing is the ability to phase genomic mutations with single-allele resolution. Phased sequencing, or genome phasing, addresses the limitation of distinguishing between variants on homologous chromosomes. Since phasing allelic compositions is crucial to understanding cancer evolution and gene expression patterns, long-read sequencing may become the standard for genotyping genes for anti-cancer drug development and patient-specific treatment.

The MinION portable long-read sequencer directly phased *EGFR* primary and secondary mutations in the lung adenocarcinoma cell line. This was a groundbreaking study, as it revealed the mutational allelic backgrounds that make tumours sensitive or resistant to anti-cancer drugs, providing useful information for determining the most effective therapeutic approaches [42]. A phasing analysis of lung cancer genomes that combined short- and long-read sequencing data produced long-phased blocks of 834 kb. The phased data revealed that cancer genomes contain regions where mutations are unequally distributed between the two haplotypes, emphasising the need for haplotype-resolved cancer genomes to track allele-specific tumour events [43]. Recently, long-read sequencing of paediatric cancer genomes using Pacbio HiFi technology identified multiple mutation changes as well as information on copy number variants, structural variants, and methylation status, all fully phased [44]. Mutations in promoters and enhancers, common in cancer cells, can cause a change in the rate of transcription, altering the level of gene expression. *TERT* transcription is often upregulated by *TERT*-promoter mutations; promoter and downstream exonic regions are usually hundreds of bases apart [45], and due to this distance, both regions cannot be captured with a single short read. Long reads provided by PacBio sequencing were used to compare *TERT*-promoter methylation patterns and gene-expression effects between wild-type and mutant cancers. This analysis revealed differences in methylation profiles and responses to demethylating agents [46]. The *PIK3CA* oncogene is one of the most frequently mutated oncogenes in all human cancers [47], and it is a typical target for cancer therapy. Somatic SNVs in *PIK3CA* are common in human breast cancer, with the majority resulting in kinase gain of function and oncogenicity. PacBio sequencing was used to phase the allelic configuration of *PIK3CA* mutations in breast cancer patients with double mutations in *PIK3CA*. Double mutations enhance PI3K signalling and promote tumour growth, yet they have a greater susceptibility to PIK3CA targeted therapy compared to single mutations. Long-read sequencing uncovered the mutational and phase status of *PIK3CA*, enabling the identification of breast cancer patients who are most likely to derive benefits from PI3K signalling targeted drugs, paving the way for personalised medicine [48].

There are not yet any FDA-approved cancer treatments that specifically target phased mutations. However, this is an area of active research, and it is possible that in the future, mutation phasing will be used to develop more personalised and effective cancer treatments.

#### 4.1.2. Characterization of Structural Variants

SVs are defined as large genomic changes of more than 1 kb [49] length, such as large indels, duplications, and inversions, or chromosomal rearrangements like translocations. However, this definition has been revised to encompass structural variants within smaller genomic regions, and SVs may be broadly defined as variations in the human genome longer than 50 base pairs [50]. Across cancers, SVs account for 55% of driver mutations (genetic mutations that drive the onset and progression of cancer), outnumbering point-driver mutations [51]. Structural variations alter the copy number of proto-oncogenes and delete or disrupt tumour suppressor genes, leading to a change in gene dosage [39]. SVs disorganise the 3D genome structure and lead to enhancer-hijacking, which is an oncogenic rearrangement of enhancers caused by the translocation or inversion of noncoding DNA regulatory regions. This ultimately results in abnormal expression patterns [52,53,54]. SVs create extrachromosomal DNA with different chromatin compaction patterns, promoting a genomic environment for oncogenic expression [55].

In 2016, a groundbreaking study demonstrated the capability of long-read technology to detect structural variants associated with cancer. The study used MinION sequencing to identify SVs in the *CDKN2A* and *SMAD4* tumour-suppressor genes of pancreatic cancer cell lines [56]. A few years later, oncogene amplifications and complex rearrangements in a breast cancer cell line were observed using PacBio sequencing. The study produced detailed maps of structural variations in a cancer genome and discovered nearly 20,000 SVs, most of which short-read sequencing had missed [57]. Similarly, Aganezov et al. applied long-read sequencing (ONT and PacBio) to breast cancer genomes and discovered hundreds of SV that were missed by NGS sequencing, emphasising the importance of LRS in cancer diagnosis and treatment [58] The development of a sensitive open-source method for SV identification enabled a benchmarking comparison between long and short reads. The results showed that long reads outperformed short reads in finding SVs [59]. Whole genome sequencing of liver cancer samples with ONT previously sequenced using short-read technology allowed the cataloguing of a comprehensive list of polymorphic and somatic SVs, as well as their potential aetiologies [60]. In a recent study, the ONT PromethION high-throughput platform was used to sequence the whole cancer genomes of 21 patients affected by colorectal cancer (CRC). The study accurately found somatic SVs in the cancer genomes. It revealed the presence of large-scale inversions of key tumour suppressor genes like *APC* and *CFTR,* which altered their expression or structure. The study also identified a new gene fusion, *RNF38*–*RAD51B*, that may facilitate the movement, invasion, and spread of CRC cells [61]. A new form of SV, heterocateny, was discovered in human papilloma virus (HPV)-positive tumors using ONT and PacBio sequencing. Heterocateny is caused by HPV integration into the host genome resulting in dysregulated recombination events of repetitive sequence and increased oncogenicity [62].

Despite the fact that long reads are more effective than short reads at identifying SVs, it has recently been proposed that multiple technologies should be combined to improve the accuracy and sensitivity of SV identification [63].

#### 4.1.3. Identification of Fusion Genes

Gene fusion in cancer cells is a category of molecular aberrations primarily caused by genomic translocations, insertions, deletions, or inversions. A substantial proportion of fusion genes drive tumorigenesis and represent target molecules with diagnostic and therapeutic potential [64]. Fusion of *RET*, *ALK*, and *ROS1*, drives tumorigenesis in lung cancers [65], while *ETV6*–*RUNX1* triggers clonal evolution and cancer progression in acute lymphoblastic leukaemia [66]. Similarly, *TMPRSS2*–*ERG* gene fusion is the most prevalent genomic alteration in prostate cancer [67] due to chromoplexy, a genetic rearrangement that is one of the key drivers of tumour evolution [3]. Gene fusions have thus served as highly specific diagnostic markers, prognostic indicators, and therapeutic targets [68].

Current clinical laboratory methods detect recurrent and novel oncogenic gene fusions using various methods and turnaround times. Fluorescence in situ hybridization (FISH) is one of the fastest methods, able to identify a single gene target in as little as 24 to 48 h for urgent clinical needs. FISH, however, can only detect one gene target per test and is insensitive to gene fusions caused by small inversions, insertions, or deletions. When multiple genes need to be investigated multiple independent tests are required. All of the limitations listed above can be overcome using newly developed clinical assays based on ONT sequencing. These assays leverage the long-read length, low cost and real-time data-acquisition capabilities of the ONT Flonge sequencing system to identify oncogenic gene fusions within a 24 h timeframe [69,70]. The fusion genes linked to leukaemia were recently identified using amplification-free CRISPR–Cas9 targeted enrichment and ONT sequencing of cell lines and patient samples. The portable MinION and Flongle ONT devices were used to bridge bedside and rapid molecular diagnostics [71].

#### 4.1.4. Whole Genome of Single Cells

Emerging roughly a decade ago, single-cell whole-genome sequencing is now an active field of research with the potential to answer fundamental questions in several areas of cell biology, including somatic mutations within individual cells, tumour evolution, and de novo mutation rates. Single-cell whole-genome sequencing approaches have been relying on Illumina short-read sequencing. No long-read applications for single-cell genomics existed until recently when a long-read protocol was introduced [72]. The method used PacBio HiFi sequencing to analyse individual human single T-cells, showcasing the feasibility of sequencing complete genomes at the single-cell level with long reads. The method still has some limitations, most of which stem from the amplification of single-cell genomes. However, advances in sequencing technologies and methods may enable the reconstruction of entire genomes of individual cells using long sequencing reads in cancer research.

#### 4.1.5. A Personalised Cancer Genome

A personalised genome assembly has long been proposed as a means of detecting all cancer somatic events. A recent study used multiple-sequencing technologies, including short reads, linked reads, and long reads (PacBio and ONT), to build the first de novo assemblies of a tumour–normal pair from the same breast cancer patient. The personalised genome assembly was compared to the standard reference GRCh38 genome assembly, revealing significant improvements in detecting somatic genetic variants [73]. The use of a personalised genome as a reference for somatic mutation calling in tumour–normal paired samples is promising, and further developments, such as reduced cost and simplified workflow, are needed for application in precision oncology.

### 4.2. Transcriptome Variation in Cancer Tissues

#### 4.2.1. Full-Length Transcriptome of Cancer Cells

Next-generation sequencing generates short RNA sequences (RNA-seq) that cover only a portion of the full-length mRNA transcript. This creates ambiguity in the alignment of short reads to isoforms and complicates full transcript analyses. In contrast to short-read sequencing, long reads can encompass the entirety of the transcript sequences and the full isoforms can be accurately determined.

PacBio Iso-seq and ONT platforms can be used for full-length complementary DNA sequencing to detect splicing isoforms and fusion transcripts. A recent study used a combination of PacBio Iso-Seq and Illumina short-read RNA sequencing to study the whole transcriptome of gastric cancer. The findings revealed numerous previously unknown transcript isoforms as a result of significant level splicing events. Those novel isoforms then could be used to provide prognostic information [74]. Another study examined the complexity of the colorectal cancer transcriptome using the same long/short reads approach and identified over 62% novel transcripts. These novel transcripts had more exons but a shorter coding sequence, were expressed at lower levels, and were probably sample-specific. In addition, oncogenes showed a substantial number of novel transcripts that may play a crucial role in carcinogenesis and tumour progression [75]. The utilisation of full-length transcriptome profiling is not yet widespread, but it holds the potential to unveil novel biological insights, biomarkers, and drug targets.

Long-read transcriptome sequencing has the ability to detect fusion transcripts, which are implicated in the development of several types of cancer due to a trans-splicing event that merges two pre-RNAs into one transcript [76]. At present, few studies have used a long-read RNA sequencing approach to detect fusion transcripts in cancers. Early studies used hybrid sequencing to correct with short-reads the high error rates of long-read sequencing [77], a method that is still used today. PacBio sequencing and Illumina RNA-seq were used in a recent study to investigate oesophageal squamous cell transcriptomics. PacBio sequencing detected five to ten times more fusion transcripts than Illumina, the majority of which were novel [78]. More recently, structural variants and fusion genes were detected in breast cancer samples through long-read genomic and transcriptomic sequencing via the ONT and PacBio platforms [79]. The utilisation of Pacbio full-length transcriptome sequencing in breast cancer cell lines revealed the presence of various new gene fusions within nested genomic variants [57]. Additionally, the analysis of transcriptome profiles in four rare cancer types using shallow ONT cDNA sequencing successfully identified distinct fusion genes [80]. This last study is particularly important because it showcases the effectiveness of ONT in profiling tumour transcriptomes with limited coverage while remaining efficient and cost-effective.

The current trend is to use only long-read sequencing and target sequencing technologies with low error rates, such as PacBio HiFi. Recently, a new fusion detection tool (pbfusion) designed specifically for Iso-Seq HiFi data was proposed and applied to sarcoma patients. The identification of known and novel fusions, as well as validated driver events, demonstrates the capability of Iso-Seq HiFi sequencing to identify fusion transcripts with absolute reliability [81].

Currently, the detection of gene fusions using conventional short-reads technology remains prevalent due to its well-established nature. Nonetheless, TGS is very promising, and while it is still being developed, it is frequently used in conjunction with short-read sequencing to achieve a more thorough analysis. Short reads are useful for detecting and validating fusions early on, whereas TGS provides critical information about the precise fusion structure and isoform involved [82]. Long-read sequencing is emerging as an efficient technique for detecting novel isoforms, fusions, and splicing events that would not otherwise be detected by short-read sequencing.

#### 4.2.2. Post-Transcriptional RNA Modifications

Post-transcriptional RNA modifications play a big role in many important cellular processes, including controlling gene expression and fine-tuning the functions of RNA molecules. To decipher what these post-transcriptional modifications do in different circumstances, it is necessary to precisely determine their transcriptomic locations and modification levels under some specific cellular conditions. It is possible to identify multiple RNA modifications with single-molecule resolution by using ONT direct RNA sequencing [18,83] which preserves modification information at the single-read level without requiring reverse transcription or PCR amplification.

Direct RNA sequencing can detect m6A patterns, a kind of post-transcriptional modification that influences the development of specific cancer types, such as glioblastoma. The capability of this sequencing protocol was recently demonstrated by detecting m6A patterns in glioblastoma cell lines [84].

#### 4.2.3. Single-Cell Transcriptomics

Full-length RNA sequencing is currently being applied at the single-cell level, offering insights into allelic and isoform variations in the transcriptome of each cell. Single-cell transcriptomics provides a thorough understanding of cancer heterogeneity and identifies key drivers of cancer progression. Single-cell transcriptomics can detect genes and pathways that are only expressed in subpopulations of cancer cells, as well as identify rare or distinct subpopulations with varying aggressiveness, metastasis ability, and treatment response [85]. This knowledge is essential for precision medicine and developing targeted treatments for various types of cancer. Several studies have used single-cell transcriptomics to identify key markers associated with drug response, enabling targeted approaches and drug-sensitivity prediction at the cellular level [86].

Currently, long-read single-cell sequencing employs droplet barcoding systems (e.g., 10× Genomics Chromium system) to barcode full-length cDNAs and sequence them on third-generation sequencing platforms [87,88]. The power of this approach is exemplified in a study of patients with metastatic ovarian cancer using PacBio single-cell sequencing. The study identified more than 150,000 isoforms, with one-third being novel. It also revealed gene fusions and alternative polyadenylation sites, providing insights into the metastatic pathway and epithelial-to-mesenchymal transition [36].

Long-read single-cell transcriptomics can be extraordinarily effective, and further cancer-related applications are anticipated.

#### 4.2.4. Cancer Epigenomics in Long-Read Sequencing

DNA methylation is an essential epigenetic modification that plays a crucial role in the regulation of numerous biological processes. Aberrant DNA methylation has been implicated in many types of cancer, affecting the cell type, state, transcriptional regulation, and genomic stability. Cancer cells can bear abnormal DNA methylation, such as genome-wide hypomethylation and site-specific hypermethylation, mainly targeting CpG islands in gene expression regulatory elements [89]. TGS technologies provide direct detection of DNA methylation with high reproducibility and low bias. One of the most common genomic modifications is 5mC, which is often found at CpG dinucleotides in the human genome. Both PacBio and ONT offer a way to detect 5mC without relying on short-read bisulfite sequencing, the traditional method of 5mC detection [23,90].

Oxford nanopore sequencing was used to simultaneously profile CpG methylation and detect somatic transposable element mobilisation in paired tumour and normal liver samples [91]. The ONT MinION platform was used to sequence the genome and epigenome of brain tumours in real time on the same day. This produced copy numbers and methylation profiles [92] showcasing a highly promising approach for categorising cancer, based on molecular markers. This has the potential to enhance clinical diagnosis and prognosis. A further example of the power of real-time ONT sequencing in a cancer setting was the accurate classification within two hours of DNA methylation profiles from surgical tumour biopsy samples assisting decision-making during live brain surgery [32].

Although nanopore sequencing is the most common long-read method for methylation analysis, the usage of PacBio is steadily increasing. One factor contributing to this is the integration of artificial intelligence with Pacbio applications. A novel deep-learning-based approach utilised PacBio sequencing to detect and determine the presence of DNA 5mC in specific genomic regions [23]. The plasma DNAs of hepatocellular carcinoma (HCC) patients were sequenced with the Illumina and PacBio platforms and the methylation at CpG sites were compared with controls. Compared to controls, HCC patients had lower overall methylation and distinctive methylation motifs. Methylation patterns on longer PacBio reads had a higher diagnostic power, compared to short reads and, in the future, could potentially be used for clinical applications [93].

TGS platforms provide genomic coverage with less GC bias, identify CpG islands at lower read depths, and enable greater experimental reproducibility in comparison to short-read methylation studies [94]. In addition, epigenetic modifications can be examined directly on native DNA without the need for PCR amplification [95].

### 4.3. Liquid Biopsy

One powerful and non-invasive method to determine the genomic status of a tumour is to use liquid biopsy techniques, such as analysing cfDNA in plasma and/or urine. Next-generation sequencers have been extensively utilised for conducting sequencing-based analyses of cell-free DNA, which offer high throughput but lack scalability and accessibility due to instrumentation costs. These challenges can be resolved by employing scalable sequencing and cost-efficient platforms, such as ONT sequencers.

Initial efforts to sequence cfDNA for non-invasive prenatal diagnosis with ONT returned unsatisfactory throughput [96]. However, due to recent technological advancements, it is now feasible to develop a protocol for efficiently sequencing cfDNA with ONT. The protocol employs low-coverage nanopore sequencing to detect copy numbers in the plasma of cancer patients. This method’s performance is similar to NGS techniques, but it is faster as it provides real-time delivery of copy number profile results [97]. A different study used ONT sequencing to specifically analyse cfDNA, providing additional proof that found that long-read sequencing can generate genomic data from liquid biopsies with a sensitivity comparable to short-read sequencing. The study used nanopore consensus sequencing on cfDNA and accurately detected *TP53* mutations at such extremely low frequencies as 0.02% [98].

The methylation level in circulating tumour DNA can be profiled using ONT sequencing, as revealed by a recent study showing methylation pattern changes specific to cancer cells as well as cancer-associated fragmentation signatures [99]. Similarly, a PacBio sequencing method for cfDNA detection and direct methylation analysis in cancer patients was recently proposed [93]. A more recent study used Oxford nanopore sequencing on cfDNA from plasma and urine to detect somatic copy-number aberrations in less than twenty-four hours, as well as sequence cfDNA fragments of various lengths. The study revealed the presence of lengthy cfDNA fragments (>300 to 8055 bp) that contained tumour-derived molecules. This finding challenged the notion that cell-free DNA solely consisted of short DNA fragments [100].

### 4.4. Data Analysis of Cancer Genomes with Long Reads

Long-read sequencing is a powerful tool for analysing cancer genomes, but it also presents unique data-analysis challenges. The challenges encompass various aspects, such as accuracy, complex base calling, computational demands, and storage capacity requirements (Table 2).

Basecalling, the computational process of translating light intensity or a raw electrical signal to a nucleotide sequence, is the initial stage of data analysis, and it is of critical importance, as almost all downstream applications depend on it. Basecalling is more complex for long reads than for short reads, with nanopore basecalling being particularly difficult due to the electric nature of the signal. In nanopore technology, the current signal level does not correspond to a single base but is most dominantly influenced by the several nucleotides that reside inside the pore at any given time, making for noisy and stochastic data [101]. PacBio detects light intensity signals, as DNA polymerase incorporates nucleotides, and the complexity comes from segmenting the fluorescence trace into pulses, converting these pulses into bases and generating a continuous long read. This process may result in noise and spurious signals. Basecalling is a thriving field in ONT, PacBio, and the research community. In order to tackle the aforementioned challenges, there has been a continuous improvement in basecalling algorithms [27,102,103] with the aim of enhancing the quality and reliability of long-read sequencing data analysis.

Long reads produce much larger datasets than short reads, and analysing these datasets is computationally demanding and requires significantly more processing power than traditional short-read methods. Existing analysis pipelines may not be capable of handling this data volume, so developing new algorithms specifically designed for long-read data analysis is critical. These algorithms should efficiently align reads, identify variants, and account for potential sequencing errors in order to improve variant-calling accuracy. Mapping long reads to a reference genome is an excellent example of this concept, as it presents unique challenges when compared to traditional short-read sequencing. Long reads, which can span thousands of bases, are more likely to contain errors, variations, and structural variants, making precise matching difficult. New algorithms have been developed specifically for long reads, such as minimap2 [104], Winnowmap2 [105] and Mapquik [106]. Furthermore, the amount of information contained in long reads can be overwhelming. Distinguishing driver mutations (critical for cancer development) from passenger mutations (with no functional impact) necessitates robust filtering and prioritisation methods. Machine-learning approaches [107,108] can provide promising solutions by allowing researchers and clinicians to filter and prioritise variants based on the predicted functional impact, as well as identify the key drivers of cancer in each patient’s unique case. Long reads, therefore, require significant computational capacity, and using high-performance computing (HPC) resources capable of providing increased processing power and memory efficiency can be an efficient solution for addressing the computational requirements associated with analysing large datasets. Cloud-based HPC solutions can also provide scalability and accessibility.

The storage of the substantial volume of data generated by long-read platforms poses a significant challenge that is difficult to resolve, as the necessity for extensive storage infrastructure may cause research institutions and healthcare providers to incur substantial expenses. This can cause a storage bottleneck which may prevent widespread adoption of long-read technology. Advanced compression algorithms specifically designed for long-read data are required to reduce the amount of data to be stored without compromising data integrity [109]. Furthermore, patient genomic data is highly sensitive, and strong security measures, such as encryption and access control protocols, are required to protect long-read data from unauthorised access or breaches.

The future of cancer genomics with long reads looks very promising, but proper clinical integration remains a bridge to build, as there are additional challenges to overcome, including error rates and sample requirements. This will be discussed further in the following section.

## 5. The Challenge of Long Read Sequencing

Long-read sequencing is a promising technology for obtaining information about the entire cancer genome, including complex genomic aberrations, transcript isoforms, epigenetic base modifications, and phase statuses. For many years, the sequencing accuracy of long-read platforms, as low as 90%, has been the primary limitation of the technologies, making the detection of point mutations extremely difficult. In recent years rapid improvements in ONT and PacBio technologies have drastically reduced the error rate and increased the base accuracy. The latest ONT Q20+ and PacBio HiFi protocols have transformed the sequencing design by producing long reads with base accuracies exceeding 99.9% [110]. The ONT Duplex protocol, a high-accuracy technique that involves sequencing both strands of DNA, was recently used to assemble a human genome with a base accuracy greater than 99.999% (Q50) and near-perfect continuity [111]. Both TGS technologies have their advantages. PacBio HiFi technology generates more accurate sequences than ONT, while ONT can produce ultra-long reads and is scalable with small portable-to-larger benchtop sequencers. The primary requirement of both technologies is the integrity of DNA/RNA molecules, a potential limitation when dealing with cancer samples. High-molecular-weight DNA and full-length RNA molecules are not always available from clinical tissues, and sometimes, it is not possible to obtain sufficient quantities. Surgical specimens and biopsies for long-term storage are generally preserved as formalin-fixed paraffin-embedded tissues, which are usually highly fragmented and damaged, limiting their use for TGS-based experiments. Biobanking of fresh frozen tissues has the potential to overcome this limitation.

Long-read data have distinct characteristics in comparison to short-reads (Table 1), necessitating tailored bioinformatic tools for quality control and downstream processing. One of the most significant challenges in bioinformatics is the development of computational tools that can effectively exploit the characteristics of long-read data. There have been numerous existing tools for long-read data, but none of them were robust enough to deal with the initial low accuracy rate of long reads. This situation, however, has improved with the introduction of new tools, such as those developed specifically for HiFi sequencing [112]. Furthermore, improved error correction methods [113,114] are constantly introduced, improving the quality and reliability of long-read-sequencing data analysis. There are also efforts to enhance the sequencing method itself through improved chemistry [27,28]. LongQC is a quality control framework specifically designed for long-read data to identify quality issues and enhance data quality for downstream analysis [115]. A new catalogue of bioinformatic tools for analysing long reads has been introduced to help researchers choose the best tool for their needs [116]. Future bioinformatics tools are user-friendly, memory efficient, and tailored to real-time analysis, which will boost long-read sequencing applications and make the technology more commonplace.

Although TGS data’s main strength is its long-read length, increasing it would benefit genome assembly and sequencing of difficult genomic regions like centromeres, telomeres, or very long complex rearrangements. When read lengths reach a certain range or cover whole chromosomes, genome assembly will be smoother, with better accuracy and completeness, and may need less computing power. Personalised cancer genomics may then be feasible and accessible, but having megabase-scale or longer reads will require developing suitable high molecular weight DNA extraction protocols and size selection methods, as well as protocols to preserve intact ultra-long DNA fragments.

Despite the diverse applications of long-read sequencing technologies, their routine application in a clinical setting faces unique challenges. Translating long-read findings into actionable insights for cancer diagnosis, prognosis, and treatment is an ongoing effort. Further research is needed to develop methods for incorporating long-read data into clinical decision-making. This will enable clinicians to tap into the potential of long-read data to develop more personalised treatment approaches. The majority of the applications mentioned in this review involved research studies. Integration of long-read sequencing into routine clinical practice is highly dependent on the stability and reliability of sequencing platforms. The flexibility presented by the portability of ONT’s MinION device in terms of working space is undeniable but integrating it into a diagnostic laboratory presents increased IT infrastructure challenges in terms of data storage, security, and protection of confidential data.

## 6. Conclusions and Future Perspectives

To summarise, long sequencing technologies hold immense potential in the field of cancer research. Despite the direct competition between the two dominant long-read technologies, the complementary use of both sequencing platforms can yield excellent results. PacBio HiFi sequencing is highly accurate, making it perfect for tasks where accuracy is essential, like SNV detection and haplotype phasing. ONT generates the longest contiguous sequence reads and allows for faster read production in real time. This enables quick characterization of SNVs during the sequencing process. The portability of ONT long-read sequencing makes it suitable in clinic-based or point-of-care settings. Over time, these technologies will become more commonplace in laboratories and clinics, as their costs continue to reduce, their accuracy improves, and their throughput increases.

The implementation of long-read technology will significantly alter the methods by which we identify and categorise mutations in cancer. Instead of aligning reads to a reference genome and inferring genetic variations, long reads will be used to assemble complete haplotypes that fully resolve complex genetic variations, either with or without a reference genome. New graph-based reference genomes representing genetic variations across a population, including large-scale structural variations like inversions and duplications, are likely to be developed [117]. Complete cancer genomes generated by long-read technologies will likely include functional information such as cellular epigenetics and transcriptomics changes. We are currently in the early phases of a revolution in the field of cancer sequencing, which is expected to have a significant and far-reaching effect.

## Figures and Tables

**Figure 1 cancers-16-01275-f001:**
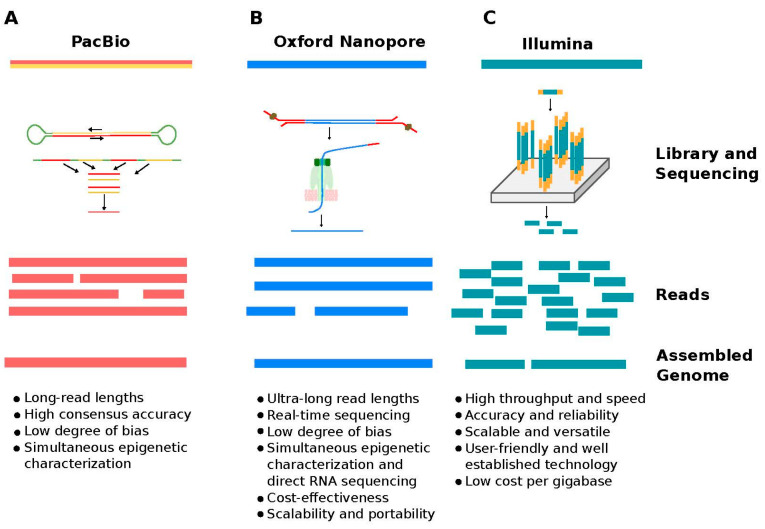
An overview of long-read and short-read methods, as well as the advantages of each technology. (**A**) PacBio HiFi sequencing technology. A DNA fragment is ligated to hairpin adapters to create a topologically circular molecule called SMRTbell. It is loaded onto an SMRT Cell for sequencing. The same DNA molecule is sequenced multiple times or passes. Every iteration produces a “read,” which may contain errors in specific bases. Circular consensus algorithms analyse and merge reads from multiple iterations to remove errors and produce a highly accurate consensus read. (**B**) ONT sequencing. DNA is tagged with sequencing adapters preloaded with a motor protein on one or both ends. A single DNA molecule is passed through a protein pore. As the molecule passes the pore, it changes the electrical current in a distinct manner, determined by each nucleotide. The disruption is measured and utilised to establish the genetic sequence. (**C**) Illumina technology. DNA is fragmented down into smaller pieces and then linked to adapters. Following library preparation, individual DNA molecules are sequenced to generate short reads.

**Figure 2 cancers-16-01275-f002:**
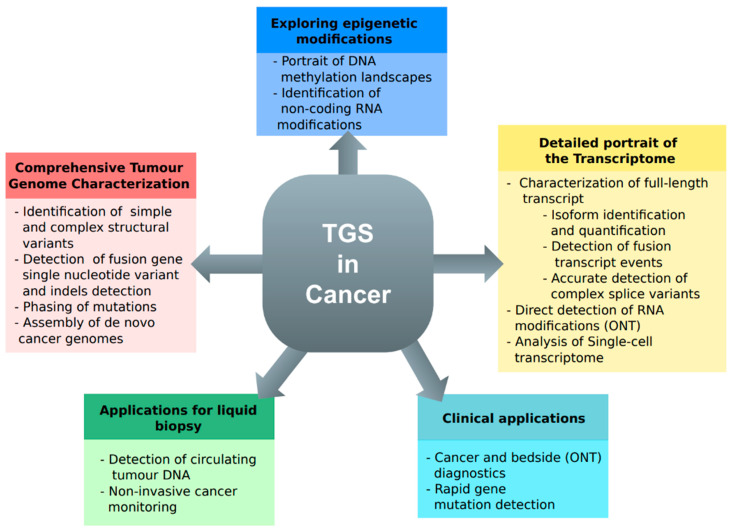
Primary applications of third-generation sequencing for investigating cancer-related inquiries.

**Table 1 cancers-16-01275-t001:** Summary of advantages and disadvantages of long and short reads technologies.

Feature	TGS	NGS
**Throughput**	Lower throughput (fewer reads but more runs)	Higher throughput (billions of reads per run)
*Advantage*	More flexibility, enables rapid sequencing of many runs	Cost-effective for sequencing many samples in a run
*Disadvantage*	Higher cost per gigabase sequenced	Fewer runs compared to TGS platforms
**Read Length**	Longer reads (10 kb–1 Mb+)	Shorter reads (150 bp–300 bp)
*Advantage*	Enables sequencing of entire transcripts and long-range variant detection	Suitable for most applications requiring moderate read lengths
*Disadvantage*	Lower accuracy as longer stretches can be more prone to errors.	Read lengths limit applications requiring long-range information
**Error Rate**	Higher error rate compared to NGS	Lower error rate compared to TGS
*Advantage*	Error rate is rapidly improving and approaching the NGS rate.	Provides high-accuracy data for most applications
*Disadvantage*	Lower accuracy compared to NGS	May require higher sequencing depth for some applications
**Cost**	Generally higher cost per gigabase	Generally lower cost per gigabase
*Advantage*	The cost has been steadily decreasing	Cost-effective for large-scale sequencing
*Disadvantage*	Costs can still be significant depending on project requirements	Costs can still be significant depending on project requirements
**Data Analysis**	Real-time analysis, portability and appropriate for whole genome assembly	Established analysis pipelines and bioinformatics tools readily available
*Advantage*	Reduced bias due to minimal amplification	Streamlined data analysis with well-established tools
*Disadvantage*	Can be computationally demanding	Data analysis can be complex for some applications
**Applications**	Ideal for large genome sequencing, de novo assembly, long-range variant detection, full-length transcriptomics, direct detection of DNA/RNA modifications, metagenomics	Wide range of applications including targeted sequencing, variant analysis, gene expression studies, and microbiome analysis
*Advantage*	Full-length transcript sequencing and accurate assembly	Versatile platform for various research areas
*Disadvantage*	Less suitable for targeted sequencing and high-depth variant analysis	May not be suitable for complex variant detection or de novo assembly
**Sample requirements**	More stringent quality and quantity requirements than NGS	Established lab workflows and less stringent requirements
*Advantage*	More stringent requirements produce long-read data	Standardised workflow and less stringent sample requirements. Degraded (FFPE) or low-input (surgical biopsy) samples can be sequenced.
*Disadvantage*	Many sample types cannot be easily sequenced due to reduced quality or small quantity.	Although more samples can be sequenced data suffers from disadvantages inherent in short-read data

**Table 2 cancers-16-01275-t002:** Summary of data complexity in TGS and NGS.

Feature	TGS	NGS
**Basecalling Complexity**	More complex due to indirect signal interpretation and longer reads	Less complex due to direct imaging and shorter reads
**Computational Analysis**	More powerful computing resources are required for assembly and variant calling	Generally less computationally demanding
**Data-File Size**	Larger files per gigabase sequenced due to longer reads	Smaller files per gigabase sequenced due to shorter reads
**Data-Analysis Challenges**	Requires specialised algorithms to handle longer reads and higher error rates	Requires robust algorithms for high-throughput data processing
**Genome Assembly**	Easier for complex or repetitive genomes due to long reads More challenging due to higher error rates and potential for chimeric reads (merged from different fragments)	More challenging for complex genomes due to shorter reads Easier due to lower error rates and shorter reads providing more overlap
**Variant Detection**	More powerful for detecting large insertions/deletions and structural variants	Well-suited for detecting single nucleotide variants

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
