# Peer review of "The Application of Long-Read Sequencing to Cancer"

_cancers, 2024, doi:10.3390/cancers16071275_

Round 1
Reviewer 1 Report
Comments and Suggestions for Authors
This is a comprehensive and timely review of third generation sequencing.
Comments on the Quality of English LanguageNone
Author Response
We are extremely grateful for the reviewer's feedback and are delighted to have received it.
The manuscript however has been improved by adding two further tables (Table 1: Summary of advantages and disadvantages of long and short reads technologies. Table 2: Summary of data complexity in TGS and NGS) and revising its content. Highlighted in yellow are major edits.
Thank you
Reviewer 2 Report
Comments and Suggestions for Authors
Specific Comments
1) Figure 1: While informative, the figure could include a comparison of the actual throughput and cost per sample for TGS versus NGS platforms to give readers a clearer understanding of the practical implications of adopting TGS.
2) Lines 218-223: The discussion on the phasing of genomic mutations could be further explained and strengthened by including specific examples of how this has changed treatment decisions or improved outcomes in cancer care.
3) Lines 398-409: The potential of single-cell TGS in revealing cancer heterogeneity is intriguing. The authors should elaborate on how this information might influence treatment strategies or contribute to the development of new therapeutic targets.
4) The manuscript is well-structured and clear, making it relevant for the field. However, the inclusion of more recent references and examples would enhance its relevance and impact.
5) The experimental design and the applicability of TGS in cancer research are scientifically sound. However, the discussion on methodological aspects lacks depth, particularly in terms of data analysis challenges and solutions.
6) Figures and tables are appropriate and support the data presented. However, additional figures comparing TGS with NGS in terms of data complexity, cost, and throughput could improve understanding.
Minor Comment: In the abstract, please correct the following sentences: "Cancer IS a multifaceted disease arising..." and "While next-generation sequencing (NGS) IS based on short..."
Comments on the Quality of English LanguageMinor editing of the English language is required.
Author Response
1) Figure 1: While informative, the figure could include a comparison of the actual throughput and cost per sample for TGS versus NGS platforms to give readers a clearer understanding of the practical implications of adopting TGS.
We greatly appreciate this feedback. The Figure is intended to provide an overview of the three methods, and a comparison of actual throughput and cost per sample would be difficult to display. The throughput of TGS and NGS is determined by several factors, including the sequencing platform used. Different TGS platforms have varying throughput capabilities, which are determined by the characteristics of each platform, as well as the characteristics of the SMRT cell used for Pacbio and the flow cell used for the ONT. The desired coverage depth can also have an impact on throughput, as higher coverage depth, which means sequencing each targeted base multiple times, improves accuracy but reduces throughput. The cost is similar. Several factors prevent the determination of a single, definitive average cost for TGS or NGS. Costs may vary greatly depending on the specific TGS or NGS platform being used. Various companies provide sequencing services with different features and prices. For example, whole genome sequencing will cost more than targeted sequencing. The cost can be influenced by the sequencing facility's overhead costs, such as labour and maintenance.
Nonetheless, we greatly value this feedback, and we have added another table to further help the reader to understand the differences between TGS and NGS. Table 1: Summary of advantages and disadvantages of long and short reads technologies.
2) Lines 218-223: The discussion on the phasing of genomic mutations could be further explained and strengthened by including specific examples of how this has changed treatment decisions or improved outcomes in cancer care.
We are grateful for the reviewer's feedback. To provide further clarity, we have added a part (lines 224-236 and lines 272-275) that discusses the role of genomic phasing-in mutations in cancer treatments and patient care.
3) Lines 398-409: The potential of single-cell TGS in revealing cancer heterogeneity is intriguing. The authors should elaborate on how this information might influence treatment strategies or contribute to the development of new therapeutic targets.
We greatly appreciate the reviewer's feedback. We have added a part (lines 433-440) that addresses single-cell cancer therapies and patient care in an effort to provide greater clarity.
4) The manuscript is well-structured and clear, making it relevant for the field. However, the inclusion of more recent references and examples would enhance its relevance and impact.
Eleven additional recent references have been incorporated: six from 2023 and four from 2022. In addition, we have included a citation from 2024 (BioArchive) regarding the high-quality genomes produced by ONT Duplex sequencing.
2024: Koren et al. 2024 [112],
2023: Van de Sande et al. 2023 [85], Ekim et al. 2023 [107], Pagès-Gallego and de Ridder 2023 [101], Zhu and Liao 2023 [115], Dorney et al. 2023 [82], Sala-Torra et al. 2023 [71]
2022: Jain et al. 2022 [106], Zheng et al. 2022 [108], Wan et al. 2022 [109], Kokot et al. 2022 [110].
5) The experimental design and the applicability of TGS in cancer research are scientifically sound. However, the discussion on methodological aspects lacks depth, particularly in terms of data analysis challenges and solutions.
The reviewer's feedback is greatly appreciated. A new section (4.4. Data Analysis of Cancer Genomes with Long Reads) (lines 512-566) has been added to provide more clarity. This section delves into the complexities of data analysis for cancer genomes, with a focus on the challenges and solutions that arise in this field.
We also updated the section "5. The Challenge of Long Read Sequencing" to include information on error rate solutions and TGS translation in clinical settings (lines 576-579, 596-599 and 614-618).
6) Figures and tables are appropriate and support the data presented. However, additional figures comparing TGS with NGS in terms of data complexity, cost, and throughput could improve understanding.
We greatly appreciate the feedback. As we stated in a previous comment, comparing TGS in terms of cost and throughput depends on a number of factors that are beyond the scope of this review; however, we acknowledge this comment and have added a table (Table 2) about data complexity.
Minor Comment: In the abstract, please correct the following sentences: "Cancer IS a multifaceted disease arising..." and "While next-generation sequencing (NGS) IS based on short..."
We apologise for this overview; we have fixed it and revised the entire abstract as suggested by another reviewer.
We are grateful to the reviewer for his/her comments which improved the quality of the manuscript. We highlighted in yellow the major edits

Reviewer 3 Report
Comments and Suggestions for Authors
The review is well organized and written, ready for publication, except some minor edits (use abbreviates afterwards) or typos (space between words).
Author Response
We are grateful for the reviewer's feedback and delighted to receive it. We have revised the whole manuscript for minor edits and revised the use of abbreviates.
The manuscript however has been improved by adding two further tables (Table 1: Summary of advantages and disadvantages of long and short reads technologies. Table 2: Summary of data complexity in TGS and NGS) and revising its content. Highlighted in yellow are major edits.
Thank you
Reviewer 4 Report
Comments and Suggestions for Authors
The authors address and interesting topic, which is likely to attract a large audience of reader. To my opinion, the manuscript needs to be seriously revised in order to achieve better clarity.
The Abstract does not deliver a message in the current form. It has to focus on advantages of long-read sequencing and present examples of breakthrough results achieved by this technology.
Although Figure 1 briefly summarizes characteristics of two long-read sequencing technologies and conventional NGS, it would better to present advantages and disadvantages of these methods in a table-like format.
The Figure 2 does not deliver important information and can be deleted.
The body of the text requires more clarity. For example, the identification of gene fusions is well established for conventional RNA next-generation sequencing, so it is not immediately clear whether long-read sequencing indeed renders and advantage. Please carefully consider the content of the entire section 1 and clearly present examples where long-read sequencing produced better results than conventional NGS.
Comments on the Quality of English LanguageMinor text editing is desirable.
Author Response
1) The Abstract does not deliver a message in the current form. It has to focus on advantages of long-read sequencing and present examples of breakthrough results achieved by this technology.
The abstract is revised to address the importance and breakthroughs of TGS in cancer genomics in response to the reviewer' comment, which is highly appreciated.
2) Although Figure 1 briefly summarizes characteristics of two long-read sequencing technologies and conventional NGS, it would better to present advantages and disadvantages of these methods in a table-like format.
We greatly appreciate this comment. Figure 1 is intended to provide an overview of the three methods, along with the advantages of each sequencing technology. Fitting a fairly complex table that represents the advantages and disadvantages of each technology would be difficult in terms of figure display and aesthetics. Nonetheless, we extremely value your feedback and have added a table (Table 1) outlining the advantages and disadvantages of TGS versus NGS. The manuscript is also updated with a second table (Table 2) summarising the complexity of data produced in TGS and NGS.
3) The Figure 2 does not deliver important information and can be deleted.
We acknowledge and appreciate the suggestion. Even if the figure does not provide critical information, we believe that including it can assist a reader who is unfamiliar with TGS. Your feedback has been greatly appreciated.
4) The body of the text requires more clarity. For example, the identification of gene fusions is well established for conventional RNA next-generation sequencing, so it is not immediately clear whether long-read sequencing indeed renders and advantage. Please carefully consider the content of the entire section 1 and clearly present examples where long-read sequencing produced better results than conventional NGS.
We appreciate the feedback provided by the reviewer. Furthermore, we recognise that it may be unclear in section 1 whether some of the TGS techniques described in this manuscript provide an advantage, which is especially true in the clinical setting, but we provide a lengthy list of examples in section 4 (The long-read approach in cancer).
Nonetheless, we provide an additional statement on the impact and promise of TGS (lines 56-60), and we expand on the impact of TGS in clinical settings throughout the manuscript. The use of third generation sequencing in clinics is still in its early stages, and the clear benefit of TGS on a clinical workflow has yet to be determined. Nonetheless, the technology is extremely promising, and we expect clinical applications in clinical diagnosis and prognosis, as demonstrated in various sections of the manuscript, particularly in relation to ONT (lines 187-189: "Using a high throughput ONT sequencer and parallel sequencing across multiple flow cells it was possible to diagnose disease-causing genetic variants in less than 8 hours for two critically ill patients" or lines 336-338 showing the identification of oncogenic gene fusions using ONT within a 24-hour timeframe). An additional and more recent example of ONT identification of fusion genes has been provided (lines 338-342) and further statements have been added to clarify the role of TGS, particularly in clinical settings and therapy: lines 272-275, and , 614-618.
We also recognise that, while gene fusions are well established in traditional RNA next-generation sequencing, they are still in their early stages in long read sequencing. We have revised the section and we clarified this in lines 409-414.
Minor text editing is desirable.
We have revised the whole manuscript for minor edits while major edits are highlighted in yellow.
We are grateful to the reviewer for his/her comments which improved the quality of the manuscript.

Round 2
Reviewer 2 Report
Comments and Suggestions for Authors
The authors have revised their manuscript and responded point-by-point to my queries. I am happy to endorse the publication of this work.
Reviewer 4 Report
Comments and Suggestions for Authors
The authors addressed the comments in a proper way
Comments on the Quality of English LanguageMinor editing required